# Unlocking Compositional Understanding of Vision-Language Models with Visualization Representation and Analysis

## Abstract

Vision-language models (VLMs) have made significant advances, debates persist about their ability to understand the combined meaning of vision and linguistic. Existing research primarily relies on computer vision knowledge and static images to deliver insights into compositional understanding of VLMs. There is still a limited understanding of how VLMs handle subtle differences between visual and linguistic information. This paper introduces an interactive visualization and analysis approach from outside the computer vision community. We found that CLIP's performance in compositional understanding only slightly exceeds the chance level of 50%. Particularly, it primarily relies on entities in visual and textual modalities, but is limited in recognizing spatial relationships, attribute ownership, and interaction relationships. Additionally, It behaves more like a "bag-of-words" model and relies on global feature alignment rather than fine-grained alignment, leading to insensitivity to subtle perturbations in text and images.

## 1    Introduction

In vision-language research, "composition understanding" involves the ability to process text and images—managing not only words, phrases and their combinations but also recognizing independent elements in images (such as objects, actions, or scenes), understanding how these elements are interrelated, and how they collectively function within a given context. For instance, the model should be able to recognize each component (such as "lawn", "girl", "white dress", "yellow ball") as well as their combination (the scene semantics of "a girl in a white dress playing with a yellow ball on the lawn").

The recently introduced CLIP Radford et al. (2021), through contrastive learning on large-scale datasets, has demonstrated remarkable capability in understanding both vision and linguistic information. Though vision-language models (VLMs) show high performance on numerous established benchmarks, however, their effectiveness in composition understanding remains a matter of debate Yuksekgonul et al. (2022). Humans can easily perceive the vision differences between images depicting "there is a mug in some grass" and "there is some grass in a mug". However, it's still unclear how well VLMs grasp the complexity of such vision-linguistic compositions. Recent studies Tejankar et al. (2021); Parcalabescu et al. (2021); Thrush et al. (2022); Diwan et al. (2022); Yuksekgonul et al. (2022) reveal even advanced VLMs struggle with challenge of integrating vision and linguistic information, especially when dealing with fine-grained linguistic phenomena.    As shown

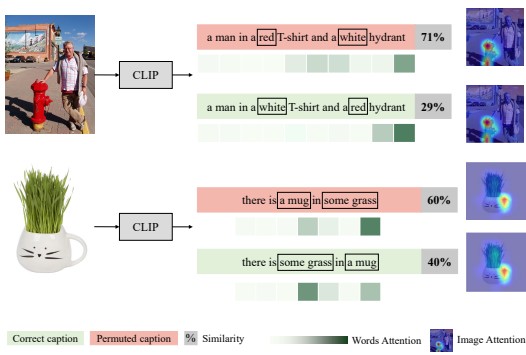

Figure 1: Two examples reveal CLIP's limitations in composition understanding. Permuted captions match better than correct ones. Visualization representation shows a tendency to focus on individual words and image objects, rather than their compositional relations.

in Fig.1, CLIP fails to match true captions.

Specifically, Winoground introduced by Thrush et al. Thrush et al. (2022) reveals that mainstream VLMs such as CLIP, UNITER Chen et al. (2020), and LXMERT Tan & Bansal (2019) fail to exceed chance level on this dataset. Yuksekgonul et al. Yuksekgonul et al. (2022) introduced the ARO benchmark (as shown in Fig.2) to assess ability of VLMs to handle relations, attributes, and order.

The studies mentioned have contributed to understanding VLMs' composition capabilities by introducing new benchmark datasets. However, we identify several critical shortcomings in presenting and conveying insights: (1) lack of interactive representation methods: current research primarily relies on static graphs to present findings, which limits our ability to dynamically explore model performance. For instance, it is challenging to discern the performance differences of a single image across different composition tasks using a bar chart. (2) limited data representation: existing research typically concentrates on quantitative metrics, failing to convey the feature encoding, transformation, and learning processes at the model level. A singular emphasis on accuracy or recall metrics fails to reveal why CLIP exhibits a higher matching degree with permuted caption.

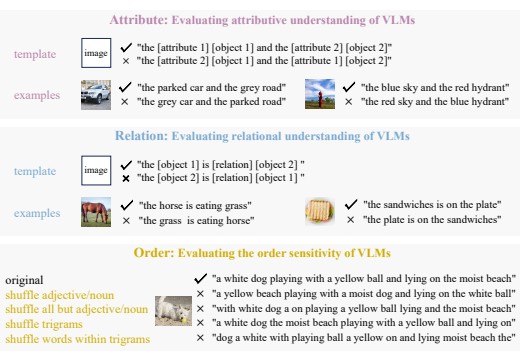

Figure 2: ARO benchmark Yuksekgonul et al. (2022) was proposed to evaluate VLMs' comprehension of relationships, attributes, and order by disrupting captions.

Our method and tool extend beyond mere reliance on quantitative metrics. It enables dynamic representation and interpretation of VLMs' behaviors in interpreting diverse vision-linguistic constructs within the pixel space, gaining the following insights: In which composition tasks do VLMs excel, and in which do they struggle? How do patterns and trends in cross-modal alignment emerge? To our knowledge, this is the first exploration of VLMs' compositional understanding from visualization representation. We fill this gap by introducing innovative representation methods and tool to interpret specific patterns where VLMs behave like bags-of-words. In summary, the contributions of this paper are as follows:

- We propose a multi-layered visualization representation and analysis method that traverses from a global overview to subspace details and down to instance specifics. This approach encompasses global grid representation of sample performance, dynamic analysis of attention biases, and interactive exploration of cross-modal alignment.

- We develop an interactive visual analysis tool that integrates cross-domain knowledge, enabling users without domain expertise to actively gain insights into VLMs' compositional understanding.

- We reveal the limitations of VLMs in compositional understanding, particularly the neglect of cross-modal fine-grained alignment. These findings will enhance the community's understanding of VLMs and provide guidance for optimization directions.

## 2 RELATED WORK AND BACKGROUND

### 2.1 VISION-LINGUISTIC COMPOSITIONITY

The recently proposed CLIP Radford et al. (2021) showcases robust joint vision-linguistic understanding. Subsequent VLMs, such as BLIP Li et al. (2022a), ALIGN Jia et al. (2021), Coca Yu et al. (2022), and Flava Singh et al. (2022), have further advanced research in this domain. These models demonstrate high performance on various benchmarks, particularly excelling in zero-shot prediction scenarios, effectively adapting to diverse downstream tasks. In vision and language research, composition understanding describes the model's capacity to identify elements in texts or images and discern their interconnected meanings. Despite matching simple images with captions seeming overly straightforward, recent NLP research Sinha et al. (2021) has shown that transformers are often remarkably insensitive to word order. Even the words in sentences are permuted, the performance of these models on downstream tasks is only slightly impacted.

Parcalabescu et al. Parcalabescu et al. (2021) assessed VLMs' ability to recognize correct linguistic phenomena in images, revealing significant challenges for current models in addressing most phenomena. Similarly, Thrush et al. Thrush et al. (2022) introduced the Winoground to evaluate VLMs' composition understanding. Mainstream models like CLIP, UNITER, and LXMERT did not surpass random chance levels despite excelling in other tasks. Their findings highlight the main challenges for VLMs: composition understanding and the integration of vision and linguistic information. Further, Yuksekgonul et al. Yuksekgonul et al. (2022) systematically assessed VLMs' ability to understand various types of relations, attributes, and order information.

## 2.2 VISUALIZATION FOR FOUNDATION MODELS

Recent strides in foundational models Bommasani et al. (2021) like BERTDevlin et al. (2018), GPT-3 Brown et al. (2020), and CLIP have surpassed our expectations. Our understanding of their internal mechanisms and how they influence outputs remains limited. Recent research indicates that visualization is pivotal in comprehending complex models Yang et al. (2023); Sacha et al. (2018).

A series of works have significantly enhanced the explainability, evaluation capabilities, and interactivity of large language models (LLM). Puchert et al. Puchert et al. (2023) used visualization techniques to hierarchically assessment LLM performance, especially uncovering "hallucinations" in knowledge subdomains. Kahng et al. Kahng et al. (2024) realized visual comparison of model outputs, revealing performance disparities on various contexts. Attention mechanism visualization research focuses on analyzing how models process sequential data, such as text or image segments, and the role of attention mechanisms in model decision-making. Yeh Yeh et al. (2023) and Li et al. Li et al. (2023b) expanded in-depth analysis boundaries of the self-attention mechanism. The former utilized joint embedding visualization of query and key vectors, providing insights into global self-attention patterns on varied input sequences. The latter offered visual exploration of attention head importance, strength, and patterns, offering insights into how ViT process image data.

Vision-language pre-trained model visualization work shifts focus towards demonstrating models' capabilities in handling cross-modal information. Recent research on VLMs interpretability addresses biases with architecture adjustment and visual technique improvements Li et al. (2023a; 2022b); Chen et al. (2022). It also introduces efficient methods for multi-object scene interpretation. Furthermore, Palit et al. Palit et al. (2023) designed the BLIP causal tracing tool, unveiling text generation mechanisms and opening new causal pathways for understanding VLMs. The interactive tool VL-InterpreT Aflalo et al. (2022) enables in-depth analysis of models' attention mechanisms and hidden states, offering new insights into vision-language interactions for understanding.

## 3 VISUALIZATION REPRESENTATION AND ANALYSIS

In this paper, we employ the CLIP for evaluation. CLIP learns representations of images and texts through the self-attention mechanism, maximizing semantic consistency between images and their corresponding textual descriptions via contrastive learning, thereby facilitating efficient cross-modal retrieval and matching. We utilize the dataset sourced from ARO benchmark Yuksekgonul et al. (2022), as shown in Fig.2. Each sample contains an image, a correct caption, and a permuted caption. Specifically, for each sample, CLIP calculates the cosine similarity between the image and captions. These similarity scores are then normalized using the Softmax function to generate the final matching scores.

### 3.1 GRID-BASED PERFORMANCE REPRESENTATION

To explore the overall performance of CLIP on the dataset, we constructed a grid representation based on cross-modal feature representations of samples (images and text captions) and their matching results, aiming to achieve the following objectives: 1) represent the overall performance of CLIP on the dataset to quantify its effectiveness and provide a global overview; 2) construct a visual proximity representation of similar semantic samples to identify CLIP's consistency and variability when handling specific permuted semantics or categories. Specifically, the pipeline is shown in Fig.3:

**Cross-modal Representation.** For each sample, we extract feature encodings of images and captions from CLIP's output, generating the cross-modal semantic representation.

**Matching Labels.** For each sample, we compute the cosine similarity between the image and both the correct and permuted captions, normalizing the scores using the softmax function to generate the final matching scores, and create three categories of match labels: *Positive Match*: The correct caption receives a higher matching score. *Negative Match*: The permuted caption receives a higher matching score. *Uncertain Match*: The matching scores for the correct and permuted caption are closely aligned (e.g., within 0.1).

**Grid Representation.** Based on the self-organizing map and resource-controlled self-organizing map Kohonen & Honkela (2007); Tu et al. (2022), we introduce homologous enhancement mechanism. Each neuron maintains a weight vector to capture sample features $w_j \in \mathbb{R}^d$ and a match vector $P_j \in \mathbb{R}^c$ to capture

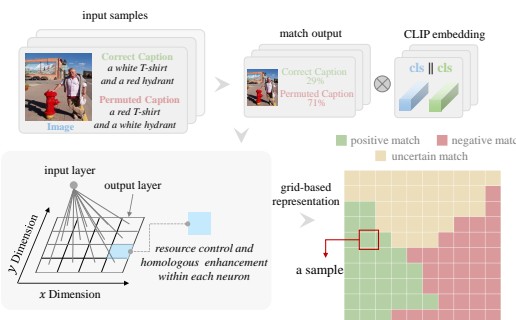

Figure 3: The pipeline of grid-based CLIP performance representation. It projects samples into a 2D topological space, visually clustering those with similar cross-modal features and matching results, providing a global overview representation of CLIP performance.

matching information, where $d$ is 1024, and $c$ is the number of matching labels. Also, each neuron is assigned a resource control right, allowing it to accept only one data sample mapping. In each iteration, the best matching unit (BMU) is selected as the neuron closest to the input sample. The BMU and its neighboring neurons are then updated in terms of their weight vectors, match vectors, and resource control rights. Ultimately, each sample is projected into a two-dimensional topological space and assigned to a single neuron, ensuring that samples with similar cross-modal features and matching results are placed close to each other. Simultaneously, samples with similar cross-modal features and matching results will be placed close to each other in the topological space, forming an organized and visually unobstructed representation.

**Visualization Presentation.** We map the topological structure to pixel space, where each cell represents a sample and is arranged in a grid according to its position in the topological space. ▮ encodes positive matches, ▮ encodes negative matches, and ▮ encodes uncertain matches. The shade of the color encodes the match score, with darker shades indicating higher scores.

### 3.2 ATTENTION-BASED SEMANTIC DIFFERENCE REPRESENTATION

To further analyze the decision-making process of the CLIP in image-text matching tasks, we employ a gradient-based attention mechanism to generate fine-grained representations. The primary objective is to reveal the regions and features that CLIP focuses on when dealing with different semantics: correct and permuted captions. The pipeline is shown in Fig.4:

**Gradient Attention Presentation.** Similarly, we input a set of image-text pairs, each consisting of an image, a correct caption, and a permuted caption, into the CLIP model to obtain the cross-modal similarity for both correct and permuted pairs. Next, for each image-text pair, we calculate the attention gradients for both image and text modalities and accumulate these across the attention layers to generate weighted attention maps. These maps highlight CLIP's focus on image patches and text tokens under different semantic conditions.

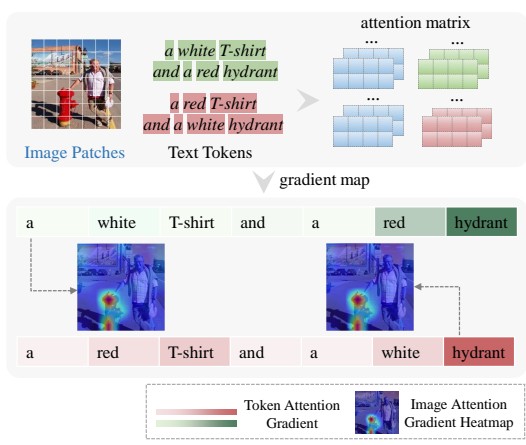

Figure 4: The pipeline of attention-based semantic difference representation. It projects CLIP's response differences to two semantic information onto image patches and text tokens, enabling direct observation of these differences.

**Visualization Presentation.** For the gradient attention representation, we employ heat maps to map it onto the original image and text sequence, offering an intuitive visual representation. Blue indicates that image patch contributes less to the matching, while orange indicates that image patch has a significant positive impact on the matching result. Similarly, the intensity of text color encodes the importance of each textual element in semantic matching. This representation can demonstrate how various visual and linguistic elements contribute to the matching outcomes. Additionally, by comparing the heat maps under two semantic conditions, we can observe the differences in CLIP's focuses and response when handling correct and permuted captions.

### 3.3 FEATURE-BASED ALIGNMENT REPRESENTATION

To explore CLIP's cross-modal alignment mechanism, we employ a collection retrieval approach based on high-dimensional feature representations to generate alignment representations. The purpose of this module is to reveal CLIP's alignment capabilities when dealing with fine-grained and compositional semantics. The pipeline is shown in Fig.5:

**Cross-modal Feature Visualization Representation.** We extract fine-grained feature representations from image patches and text tokens, including global CLS features. We then employ t-SNE Van der Maaten & Hinton (2008) technology to project these cross-modal features into a two-dimensional space, ensuring that features similar in high-dimensional space remain close in the two-dimensional plane. In terms of visualization, we map the projected features onto pixel space and use bubble sets to enhance the visual representation of different modal patches or token sets. This representation facilitates the observation of alignment within and between modalities, providing an intuitive validation of alignment.

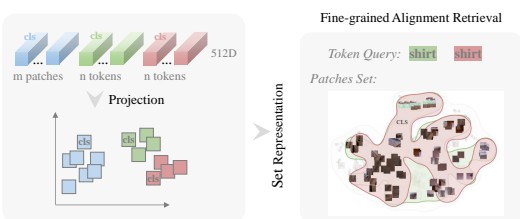

Figure 5: The pipeline of feature-based alignment representation. It projects CLIP's handling of individual and composite text semantics into an interactive pixel space, facilitating the observation of its alignment mechanisms.

**Cross-modal and Fine-grained Retrieval.** This module allows the use of text tokens as query keywords to search through collections of image patches. The retrieval process returns the top k image patches that best match the query token, based on rankings derived from cosine similarity. By observing changes within the collection, we can understand the differences in how the CLIP performs with individual text entities and composite semantics.

## 4 CASE STUDY

In this section, we present preliminary insights gained from the visualization representations and our interactive analysis tool. We randomly selected 900, 1600, and 10,000 samples (Fig.6-A1) from the dataset to observe the visualization output of the grid representation module, respectively. The differences in the number of red and green cells across these grid views are not significant. Moreover, the statistical bar chart (Fig.6-A2) indicates that CLIP's success rate in positive matching fluctuates around 50%, failing to reach the expected high level. This highlights CLIP's limitations in handling semantically permuted yet textually similar inputs. Specifically, CLIP shows insensitivity to spatial perception (e.g., "to the left of", "to the right of", "on"), spatially oriented action relationships (e.g., "sitting on the top of", "sitting on", "standing in front of"), inter-object interactions, and the object properties, all of which do not achieve upstream levels of positive matching.

Next, we randomly selected 2,500 samples from the relationship perturbations dataset for analysis, observing the performance of data subsets under different relational perturbations in the grid representation view (Fig.6-B). Sample 23371 exhibited a deep red cell, indicating a high level of negative matching. By analyzing the gradient attention differences (Fig.6-B1) for this sample under both correct and permuted caption, we found that the responses of image patches and text tokens did not change significantly due to the relationship perturbations. Conversely, the image and text heat maps were almost identical, focusing predominantly on the visual and textual entities related to "skis",

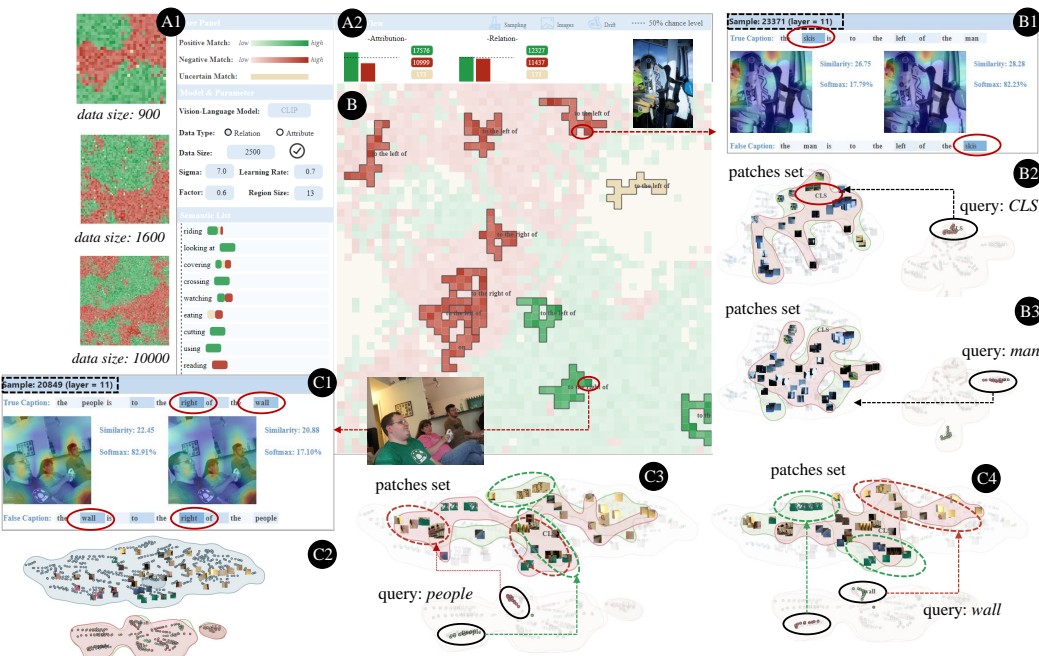

Figure 6: Case studies based on three visualization representation modules and our interactive analysis tool. We found that CLIP primarily relies on entities within the visual and textual modalities, showing limited sensitivity to relationships between these entities. This is largely due to CLIP's cross-modal alignment not adequately considering the alignment between fine-grained image patches and text tokens.

which occupied the majority of the pixel area, suggesting it was a key factor influencing CLIP's image-text matching. Similar observations were noted in other samples, suggesting that CLIP tends to focus on individual entities rather than complex semantic relationships. When querying with token "man" from both the correct and permuted semantics (Fig.6-B3), image patches sets primarily consisted of patches related to the scene, with minimal variation among the elements.

In addition, sample 20849 exhibited a deep green cell, but upon analyzing the gradient attention distribution under both correct and permuted captions, no significant differences were observed (Fig.6-C1). The image response continued to center on visual entity "people" despite relatively more attention given to the text token "right". We explored sample 20849 in the cross-modal alignment representation view. Querying with the token "people" (Fig.6-C3) yielded an image set that did not feature the expected numerous body image tokens; instead, it primarily consisted of patches related to the background wall. When querying with the token "wall" (Fig.6-C4), some image patches associated with walls were retrieved under permuted conditions, while the correct image set contained many patches related to T-shirts. These observations indicate that CLIP's cross-modal alignment is not optimal, failing to effectively focus on the alignment between fine-grained image patches and text tokens. This could be one reason for its behavior resembling that of a "bag-of-words" model.

## 5    DISCUSSION

**Our work is based on new problems and demands, providing solutions from beyond the computer vision and NLP communities.** Recent debates have questioned the composition understanding capabilities of VLMs. Additionally, as shown in Fig. 7, our user survey (20 male and 16 female participants, from fields like multi-modal image-text retrieval, multi-modal sentiment analysis, object detection and tracking, natural language processing, and visualization) found that many participants experienced poor performance in composition understanding when using popular models like CLIP, BLIP, and ALIGN. Notably, 5.6% (Q4) of participants frequently encountered these is-

sues. Existing research primarily relies on static images and domain-specific knowledge to present findings and insights, often overlooking the importance of dynamic analysis and multidimensional exploration. For instance, understanding these insights requires VLMs knowledge, such as attention mechanisms, contrastive learning, embedding techniques, and neural network architectures. Moreover, static charts only offer a limited summary of metrics, insufficient to reveal model's specific performance when processing different vision or linguistic structures. We propose an interactive visual analysis method. This approach, spanning from global overview to data subspaces and specific instances, lowers technical barriers, allowing a wider users to access and understand this topic. According to questionnaire results, 89% (Q6) participants prefer visualization methods to analyze this issue, and 92% (Q7) look forward to the introduction of related visual analysis tool.

**Our efforts bridge the gap between computer vision and visualization, fostering interest and exploration across various fields.** Our preliminary survey reveals that while many researchers use VLMs, only 13.9% (Q3) of participants recognize their limitations in integrating vision and linguistic information. This shortcoming could hinder VLMs' application in fields like healthcare, security surveillance, and autonomous driving, where a low tolerance for errors is crucial. The survey also reveals that 90% (Q7) participants believe that enhancing this capability could advance related technologies and impact multi-modal applications. Further, our study employs an intuitive approach, encouraging interdisciplinary researchers to explore deficiencies in composition understanding and to initiate discussions about unknown issues. Specifically, our research provides guidance for researchers looking to optimize VLMs.

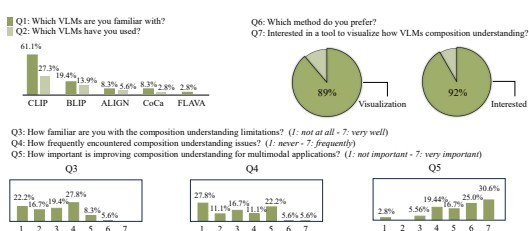

Figure 7: Preliminary user survey Q1-Q7 ($N = 36$). Results reveal 72.2% participants experience problems with composition understanding when using mainstream VLMs, with 5.6% frequently encountering these issues. However, only 13.9% are well-informed about these problems. Additionally, 89% prefer using visualization methods for analysis, and 92% look forward to the introduction of relevant visual analysis tool.

For instance, incorporating more negative samples in model training for contrastive learning, or designing network layers that more effectively learn cross-modal features, can enhance the processing capabilities of VLMs.

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
