# OpenReview forum: "Unlocking Compositional Understanding of Vision-Language Models with Visualization Representation and Analysis"
_ICLR.cc/2025/Conference — ICLR 2025 Conference Withdrawn Submission_

### Official Review · Reviewer_TVCL · 2024-10-27

**Soundness:** 1
**Presentation:** 2
**Contribution:** 1
**Rating:** 3
**Confidence:** 5

**Summary:**

This paper visualizes and analyzes the CLIP for the limitations of compositional understanding.

**Strengths:**

This paper discusses an interesting research problem.

**Weaknesses:**

However, I still have several concerns.
1.  There are many vision-language models coming out recently, e.g., GPT-4V, Gemini. Before that, there are LLaVA and its variances. Simply researching CLIP may not provide up-to-date insights.
2.  The literature review is out-of-date and some parts are missing. More recent works about compositionality should be included. Analysis works should also be included.
In summary, I think the current version is not ready for a conference paper.

**Questions:**

See weakness.

---

### Official Review · Reviewer_oZGQ · 2024-11-04

**Soundness:** 3
**Presentation:** 3
**Contribution:** 2
**Rating:** 5
**Confidence:** 2

**Summary:**

This paper presents a multi-layered visualization approach and analysis methods that move from a global overview to subspace details and down to individual instances. Using methods such as grid-based performance representation, attention-based semantic difference representation, and feature-based alignment representation, the authors have developed an interactive visual analysis tool. This tool integrates cross-domain knowledge, allowing users without domain expertise to gain insights into VLMs' compositional understanding actively.

**Strengths:**

1. The paper is exceptionally well-written, featuring impressive graphics that effectively illustrate the concepts. Ideas are clearly articulated and supported with numerous examples, enhancing comprehension.

2. The concepts are straightforward and easy to grasp, with clear interpretations provided for each example.

3. The three representation methods are innovative, offering a well-explained approach to interpreting VLMs' compositional understanding capabilities and limitations.

**Weaknesses:**

1. Given the depth of analysis, more nuanced conclusions would strengthen the paper. General statements like "This highlights CLIP’s limitations in handling semantically permuted yet textually similar inputs" and "CLIP tends to focus on individual entities rather than complex semantic relationships" are insights that can already be inferred from accuracy-based analyses alone. Such shallow conclusions could limit the paper's contribution in precisely identifying specific issues.

2. Representation clarity: The first item in the legend of Figure 4 is somewhat confusing. It appears that the red line refers to "Token Attention" and the green line to "Gradient." A simple fix would be to ensure "Token Attention Gradient" appears on the same line for better clarity.

**Questions:**

Would a universal solution to the problem of compositional understanding be possible? Alternatively, are there specific directions worth exploring to address this challenge? Discussing potential solutions or promising approaches could enhance the paper's contribution.

---

### Official Review · Reviewer_5QhM · 2024-11-05

**Soundness:** 2
**Presentation:** 1
**Contribution:** 1
**Rating:** 1
**Confidence:** 5

**Summary:**

This paper proposes a visualization system for analysis CLIP. They combine grid-based performance representations, attention-based semantic difference representations, and feature-based alignment representations, to a unified system for visualization.

**Strengths:**

The visualization looks cool.

**Weaknesses:**

Overall, this paper lacks technical novelty. All the techniques, approaches, and even the datasets are derived from existing works. The primary contribution is the engineering effort to integrate these elements into a visualization system. While this paper might be valuable for a demo paper, the contribution is not enough for research paper.

Some sections require further elaboration. For instance, the concepts of self-organizing maps and resource-controlled self-organizing maps are not clearly explained. Providing additional background and details would be better. Additionally, the motivation for using a grid representation is unclear. The authors should provide more intuition to this design choice and its implications.

Although the paper presents conclusions based on their analysis, these conclusions have already been reported in the ARO paper and other existing works. From my perspective, this paper merely offers an alternative way to show the same findings found in the ARO paper. None of new evaluation, benchmarks, techniques are proposed for a research contribution.

**Questions:**

Please see above.

---

### Official Review · Reviewer_djMT · 2024-11-06

**Soundness:** 2
**Presentation:** 2
**Contribution:** 2
**Rating:** 3
**Confidence:** 4

**Summary:**

This paper introduces an interactive visualization and analysis approach for exploring compositional understanding within Vision-Language Models (VLMs). The authors propose a multi-faceted visualization representation consisting of grid-based performance representation, attention-based semantic difference representation, and feature-based alignment representation. Additionally, the paper presents survey results indicating community interest in and demand for this work, underscoring the relevance of the study.

**Strengths:**

- **Novelty in Addressing New Needs**: This work identifies and addresses emerging problems by presenting a solution that extends beyond traditional computer vision and NLP methods.
- **Multi-layered Visualization and Analysis**: The paper introduces a comprehensive visualization and analysis approach, offering insights from a global overview to subspace and instance-specific details.
- **Interactive Tool**: The development of an interactive visual analysis tool that integrates cross-domain knowledge is useful for the community, potentially addressing a gap in compositional understanding of VLMs.

**Weaknesses:**

- **TL;DR**: This paper may be more suitable for a demo track at another conference after further improvements.
- **Limited Originality in Methods**: The methods used, such as grid-based searches and t-SNE, are well-known and may lack the novelty expected in this context.
- **Insufficient Experimental Details**: The methodology could be better detailed; for instance, in line 171, the threshold selection for matching scores is mentioned but not elaborated upon.
- **Lack of Quantitative Results**: The paper relies mainly on case studies without quantitative analysis, which may limit the strength of its claims.
- **Clarity in Writing**: Certain sections would benefit from clearer structure and transitions, and some subjective terms, like "significant" (e.g., line 257), would be clearer with statistical evidence (e.g., p-values).
- **Inconsistent Formatting**: Symbols and citations need standardization, such as in line 028 with “composition understanding,” to improve readability.

**Questions:**

1. Is Figure 6 intended as a demonstration page? If so, will the interactive tool be made available online?
2. In the literature review, there is no apparent connection between prior work and the proposed approach. Could you clarify any key relationships or distinctions?
3. The study references a survey with 36 participants. Could the authors discuss why this sample size is adequate for supporting the claims made?

---

### Official Review · Reviewer_y3Pc · 2024-11-06

**Soundness:** 4
**Presentation:** 4
**Contribution:** 3
**Rating:** 8
**Confidence:** 4

**Summary:**

The paper addresses the existence of a compositional approach in vision language models. The authors identify the presence of structure in the linguistic data associated with images and lackthereof in the image recognition algorithms. They subsequently propose a method to over the problem and test it on existing benchmarks reporting better performance.

**Strengths:**

Identification and addressing of a foundational problem, proposing a solution and better experimental result.

Further, the paper is well written and references and examples and illustrations are provided.

**Weaknesses:**

It would be good to review and reference (if you see appropriate) the literature on compositional models of language. Some of these have led to work on compositional VLM, e.g. see a recent paper https://aclanthology.org/2024.alvr-1.17/ which address a similar problem. Your approach differs from theirs, and this is good. But it would be good to have other solutions referenced in your paper.

**Questions:**

listed in  the weakness section.

---

### Note · Authors · 2024-11-19

I have read and agree with the venue's withdrawal policy on behalf of myself and my co-authors.